# Effect of Curcumin and Coenzyme Q10 Alone and in Combination on Learning and Memory in an Animal Model of Alzheimer’s Disease

**DOI:** 10.3390/biomedicines11051422

**Published:** 2023-05-11

**Authors:** Pramod Kumar, Aarti Singh, Anurag Kumar, Rahul Kumar, Rishi Pal, Amod Kumar Sachan, Rakesh Kumar Dixit, Rajendra Nath

**Affiliations:** 1Department of Pharmacology &Therapeutics King George’s Medical University, Lucknow 226003, Uttar Pradesh, India; pramodaditya650@gmail.com (P.K.); aartisingh25121994@gmail.com (A.S.); kganurag123@gmail.com (A.K.); rahulkgmu@gmail.com (R.K.); rishipalkgmc@gmail.com (R.P.); dramodkumarsachan@gmail.com (A.K.S.); dixitkumarrakesh@gmail.com (R.K.D.); 2Department of Pharmacology, King George Medical University, Lucknow 226003, Uttar Pradesh, India

**Keywords:** Alzheimer’s disease, curcumin, coenzyme Q10 dementia, neurodegenerative disorders, cognition, neuroprotectant, oxidative phosphorylation, hyperphosphorylated tau

## Abstract

The most frequent neurodegenerative illness among senior people and the main cause of dementia is Alzheimer’s disease. The present dementia medications available only help with the symptoms of cognitive deficits and have several negative effects. The current study’s goal is to assess the effects of curcumin and coenzyme Q10, two herbal medicines, both separately and in combination, on learning and memory before comparing them to the industry standard drug. A total of 42 adult healthy Wistar rats were used in our study. In this experiment, rats were given daily doses of 2.5 mg/kg of body weight of scopolamine hydrobromide for 7 days to induce Alzheimer’s disease. On the eighth day, behavioural testing was conducted. Following testing, scopolamine and the test medications were given daily for the following 21 days. On days 29 and 30, behavioural testing was conducted once more, and then animals were slaughtered. Brain homogenate was produced for the estimation of molecular and biochemical markers. Curcumin has demonstrated a dose–response relationship, with a higher dose (200 mg/kg b.w. p.o.) being more effective than a lower dose (100 mg/kg b.w. p.o.). Similar to the greater dose of curcumin, coenzyme Q10 (200 mg/kg b.w. p.o.) has also been found to improve memory and learning. Higher doses of curcumin and coenzyme Q10 had more pronounced and meaningful effects. Acetylcholinesterase and TNF levels increased in scopolamine-induced memory impairment, but these effects were restored by the test medications, and improved by the combined therapy. These outcomes are comparable to those of the common medication memantine. As a result, we may infer from our results that curcumin at higher doses and its combination with coenzyme Q10 (200 mg/kg b.w. p.o.) have a significant impact on cognitive impairment in animal models of Alzheimer’s disease and can be utilised alone or as an add-on therapy for the condition.

## 1. Introduction

A degenerative neurologic ailment called Alzheimer’s disease (AD), resulting in the death of brain cells and brain shrinkage, is characterized by a steady decline in daily living skills and cognitive function, which might result in behavioural changes. The most frequent neurodegenerative disease among senior people and the main cause of dementia is Alzheimer’s disease. Memory, thinking, orientation, comprehension, calculation, learning capacity, language, and judgement are among the multiple higher cortical functions that are disrupted by dementia, a syndrome brought on by brain diseases that are typically chronic or progressive and that might cause disability in later life [1]. According to World Health Organization (WHO) research, this condition affects 35.6 million individuals globally, and as the average life expectancy of the older population rises, it is predicted that this number will double by 2030 and treble by 2050 [2]. The characteristic pathogenic abnormalities are extracellular amyloid protein deposits, sometimes called senile plaques, and intracellular neurofibrillary tangles (NFTs). The brain parenchymal and the cerebral blood vessels exhibit amyloid plaque accumulations, commonly known as congophilic angiopathy or cerebral amyloid angiopathy. NFTs produce paired helical filaments using tau proteins that have been hyperphosphorylated. These NFTs are characterized by a variety of pathologies, including neuronal and synaptic loss [3]. A rising amount of evidence suggests that oxidative stress, neuroinflammation, mitochondrial dysfunction, and autophagy are likely the origins of Alzheimer’s disease [4].

### 1.1. Oxidative Stress and Mitochondrial Dysfunction

The body’s increased susceptibility to free radical damage is a major aspect of the cognitive aging process [5]. Free radical generation, if left unchecked, can harm cellular structures, particularly those in the brain’s nervous system [6]. Mitochondrial respiration contributes significantly to the growth of free radicals, turning 2% to 5% of all oxygen into ROS (Golden et al., 2002; Simpson et al., 2015) [5,7]. In the brain, chemical species such as hydrogen peroxide (H2O2), hydroxyl free radical (•OH), superoxide anion (O2•), and peroxynitrate (ONO2) can harm DNA, proteins, and membrane lipids, killing critical neurological structures. Antioxidants and systems that remove damaged molecules help the brain fight itself against the harm caused by free radicals. Cognitive function impairments and elevated levels of oxidative stress co-occur in AD patients [8]. Therefore, knowing the molecular mechanism through which oxidative stress can reduce cognitive function is essential for developing strategies to counteract the sneaky effects of dementia and cognitive decline [9].

Dementia is characterised as a significant decline in cognitive performance that is severe enough to affect social interaction. There are numerous varieties of dementia, each with a unique aetiology and set of signs and symptoms. The most frequent cause of dementia is Alzheimer’s disease (AD), where 60–70% of patients exhibit dementia brought on by the accumulation of beta-amyloid plaques in the brain. Learning is the primary process by which we develop our attitudes, emotional expression, adaptive, cognitive, and affective behaviours, as well as other skills [10].

The process through which sensory information is altered, condensed, elaborated, recovered, and put to use is referred to as cognition. In today’s demanding world, memory difficulties, such as poor recall, low retention, and slowing recall, are frequent [11]. The existing medications for dementia only help with the symptoms of cognitive deficits.

The first-line drugs for this are (1) Cholinergic activators/Cholinesterase inhibitors, such as rivastigmine, galantamine, and donepezil, and (2) Memantine, a glutamate (NMDA) antagonist. However, they also come with a number of undesirable side effects, such as nausea, diarrhoea, and appetite loss. From this point forward, research is concentrated on a variety of therapeutic plants that are utilized in conventional treatment and have few negative effects [12]. The need of the time is to search for new lead compounds from phytoconstituents of a plant which are equal in potency and efficacy to the present drugs used for various neurodegenerative diseases but have fewer side effects.

The herb *Curcuma longa*, generally known as turmeric, is a member of the zingiberaceae family of plants. Curcumin is an active hydrophobic polyphenol that is isolated from the rhizomes of this plant. In India and China, curcumin has historically been utilised as a treatment for a variety of illnesses [13]. Curcumin has a wide range of biological and pharmacological effects, including those that are antibacterial, anti-inflammatory, antioxidant, anti-tumour, anti-protozoa (Leis mania amazonensis), and anti-HIV. This has been demonstrated by modern medicine. These chemicals fall under the category of curcuminoids, which are similar to diarylheptanoids. According to the Joint Nations and World Health Organization Expert Committee on Food Additives (JECFA), curcumin is regarded as a safe chemical and is hence appropriate for everyday dietary usage. Curcumin is a viable drug candidate for the treatment of complicated disorders such as Alzheimer’s disease and its related cognitive loss because of its pleiotropic effects and good safety profile [14].

### 1.2. Effects of Curcumin on Macrophages

When compared to AD patients whose macrophages were not treated with curcumin, those whose macrophages were treated demonstrated improved absorption and swallowing of the plaques. Curcumin may therefore aid the immune system in eliminating the amyloid protein [15].

### 1.3. Curcumin as an Anti-Inflammatory in Alzheimer’s

In THP-1 monocytic cells, curcumin prevents the production of the Egr-1 protein and Egr-1 DNA-binding activity. Studies have demonstrated Egr-1’s contribution to the amyloid peptide-induced induction of the cytochemokine gene in monocytes. Curcumin lowers the inflammation via inhibiting Egr-1’s DNA-binding function. Curcumin can reduce the chemotaxis of monocytes, which can happen in response to chemokines from active microglia and astrocytes in the brain [16].

### 1.4. Oral Bioavailability

Curcumin has poor bioavailability. Because curcumin is readily conjugated in the intestine and liver to form curcumin glucuronides [17], absorption appears to be better with food.

NSAIDs, reserpine, and blood thinners are a few of the medicines that are said to interact with curcumin. The bioavailability of curcumin was significantly increased by 2000% when 20 mg of piperine, which is extracted from black pepper, was also taken as a supplement [18].

Alzheimer’s disease, Parkinson’s disease, and other neurodegenerative disorders were found to have defects in the inner mitochondrial membrane and oxidative phosphorylation.

Coenzyme Q10 is an endogenous proenzyme found in the inner mitochondrial membrane of human cells; therefore, it may also be a promising neuroprotectant. Coenzyme Q10 functions as an antioxidant, halting the oxidation of lipids within the mitochondrial membrane and other damage brought on by free radicals. CoQ10 is an essential cofactor of the electron transport chain, and an important antioxidant at both mitochondrial and lipid membranes [19].

According to research conducted on animals, coenzyme Q10 supplementation has positive anti-oxidative effects, enhances mitochondrial activity, and guards against ATP depletion [20]. According to a study by Ishrat et al. from 2006, coenzyme Q10 might be useful in the treatment of dementia. In animal research, oral coenzyme Q10 supplementation boosts the levels of brain mitochondria [21] and enhances cognition while lowering oxidative damage indicators in the cerebral cortex and hippocampus of rats with generated oxidative injuries.

## 2. Materials and Methods

### 2.1. Animals

In total, 42 mature female Wistar rats in good health weighing between 160 and 200 g (*n* = 6 per group) were used in this study. The animals were housed in a CPCSEA-certified institutional animal facility at King George’s Medical University in Lucknow, where they were allowed access to food and drink at will and kept in a temperature-controlled environment (25 °C). Before the trial began, a minimum of 7 days of acclimatization time was permitted, and a veterinarian routinely checked on the animals’ health. The Institutional Animal Ethics Committee (IAEC) of King George’s Medical University (K.G.M.U.), Lucknow, gave its clearance before the study could begin (Ref. No 132/IAEC/2020). All experiments in the study were conducted as per the guidelines laid down by the Committee for the Purpose of Control and Supervision of Experiments on Animals (CPCSEA). Animals were obtained from IITR, Lucknow.

The study included a total of 42 adult, healthy Wistar rats (160–200 g, *n* = 6 each group) of either sex. Animals were acclimated to the environment for seven days before to the trial, during which time they received a regular pellet feed, unlimited access to water, and 12 h cycles of light and darkness.

Just before administration, the desired drug doses were calculated based on body weight, dissolved in normal saline, and placed into a syringe with a 16 G ball-tipped oral feeding cannula. The ball-tipped oral feeding cannula was inserted from the side of the mouth into the pharynx and then released into the oesophagus while the rat was lying on the left palm. Rats’ oesophagus was then given an injection of the medication. After the animals had been properly restrained, an intraperitoneal injection was administered using a 26G needle that was placed into the right lower quadrant of the abdomen.

### 2.2. Development of Alzheimer’s Disease Model in Animals

In this experiment, rats were given scopolamine hydrobromide 2.5 mg/kg i.p. at a volume of 5 mL/kg in the vehicle in order to induce Alzheimer’s disease. It was given every day for seven days. On the eighth day, behavioural testing was conducted. Following testing, scopolamine and the test medication were administered daily for the following 21 days. Animals in each group were observed for 30 min following the test drug administration, and behavioural evaluations were conducted on the 29th and 30th days following the introduction of scopolamine. Animals in each group were sacrificed by decapitation under anaesthesia following behavioural experiments. Their brain was removed, and a brain homogenate was created for assessments of biochemical and molecular markers.

### 2.3. Drugs and Chemicals

Scopolamine hydro bromide, curcumin, coenzyme Q10, Tri HCl buffer, RIPA buffer, Protease Inhibitor, PBS, Thiobarbituric acid (TBA), and pyrogallol were purchased from Sigma chemical company, USA. Memantine and all other used chemicals were purchased from TCI, Japan. ELISA kits for TNF-α and Aβ42 estimation was purchased from Real Gene, USA.

### 2.4. Experimental Design

A total of 42 rats were divided into 7 groups (Figure 1 and Table 1). Each group contained 6 rats.

**Group 01**: Control (Vehicle)

**Group 02**: Rats were administered Scopolamine hydrochloride (2.5 mg/kg b.w. i.p.) [22]

**Group 03**: Rats were administered Scopolamine (2.5 mg/kg b.w. i.p.) and Curcumin (100 mg/kg b.w. p.o.) [23]

**Group 04**: Rats were administered Scopolamine (2.5 mg/kg b.w. i.p.) and Curcumin (200 mg/kg b.w. p.o.) [24]

**Group 05**: Rats were administered Scopolamine (2.5 mg/kg b.w. i.p.) Additionally, CoQ10 (200 mg/kg b.w. p.o.) [25]

**Group 06**: Rats were administered Scopolamine (2.5 mg/kg b.w. i.p.), Curcumin (200 mg/kg b.w. p.o.) and CoQ10 (200 mg/kg b.w. p.o.)

**Group 07**: Rats were administered Scopolamine (2.5 mg/kg b.w. i.p.) and standard drug Memantine (10 mg/kg b.w. i.p.) [26]

### 2.5. Pharmacological Parameters

Behavioural tests were performed on day 8 and then day 29 and day 30. To avoid stress to the animal, they were maintained in a quiet room for 1 h before the tests.

#### 2.5.1. Y-Maze Test

Spontaneous alternation performance was tested as described previously (Ohno et al., 2004 [27]. Each rat was placed in the centre of the symmetrical Y maze and was allowed to explore freely through the maze during an 8 min session. The sequence and the total number of arms entered were recorded. Percentage alternation is as follows: several triads containing entries into all three arms/maximum possible alternations (the total number of arms entered − 2) × 100 [27].

#### 2.5.2. Morris Water Maze Test

The animals performed 2 trials/day for 2 days (29th and 30th), with the platform submerged in quadrant 2. The time limit was 120s/trial, with a 20 min intertrial interval. Time taken to locate the platform in the acquisition phase was analysed [28].

#### 2.5.3. Elevated Plus Maze Test

The rats were each placed at the end of an open arm, away from the platform, and the time it took for them to move into either of the enclosed arms was timed (transfer latency, or TL). TL was the amount of time that elapsed from the moment the animal was positioned in the open arm until all of its legs had crossed the fine white line in the centre of the enclosed arm [29].

### 2.6. Brain Homogenate Preparation

Whole-brain samples were rinsed with ice-cold saline (0.9% NaCl) and homogenized in chilled tissue lysis buffer (RIPA + PI cocktail). The homogenates were centrifuged at 800 g for 5 min at 4 °C to separate the nuclear debris. Afterward, the lysate was centrifuged at 14,000 g for 30 min at 4 °C to obtain supernatant, which was further used for the estimation of biochemical, inflammatory, and molecular parameters.

### 2.7. Biochemical Parameters

#### 2.7.1. Colorimetric Determination of Acetylcholinesterase Activity—Ellman’s Assay

A photometric method for determining acetylcholinesterase activity of brain homogenate has been described. The enzyme activity is measured by following the increase in yellow colour produced from thiocholine when it reacts with the dithiobisnitrobenzoate ion [30].

#### 2.7.2. Superoxide Dismutase (SOD) Activity in Brain Homogenate

SOD activity in brain tissue homogenate was analysed by using the method of Marklund and Marklund and expressed in U/mg of protein. In this method, superoxide dismutase inhibits pyrogallol autoxidation. It was investigated in the presence of EDTA in the pH range 7.9–10.6. The activity was measured at 420 nm using a spectrophotometer. Protein estimation was performed by Lowry’s method.

#### 2.7.3. TNF-α Levels and Aβ42 Levels

The tumour necrosis factor-alpha (TNF-α) and Aβ42 levels estimation was performed as per the instructions provided in the enzyme-linked immunosorbent assay (ELISA) kit in brain homogenate. The type of ELISA used was sandwich ELISA. The assay was performed at room temperature.

Seven wells were taken for standard and one for blank. An amount of 100 μL each of the dilutions of standard, blank, and samples was added into the appropriate wells. Incubation was carried out for 2 h at 37 °C. After that, washing was performed and then 100 μL of biotinylated antibody was added and incubation was carried out for 1 h at 37 °C. Washing was performed again and then 100 μL of streptavidin-HRP was added and incubation was carried out for 1 h at 37 °C. Washing was repeated and then 90 μL of TMB substrate was added, the solution turned blue and incubation was carried out for 20 min at 37 °C. Finally, stop reagent 50 μL was added, the solution turned yellow, and the reading (optical density, OD) was taken by microplate reader set to 450 nm.

## 3. Statistical Analysis

The data obtained are expressed as mean ± SD and analysed using single-factor ANOVA. Further statistical analysis for individual groups was carried out by Tukey HSD post hoc test. The criterion for statistical significance is taken as *p* < 0.05 in all statistical evaluations.

## 4. Results

### 4.1. Behavioral Assessments

#### 4.1.1. Effect of Curcumin and Coenzyme Q10 on Transfer Latency in the Elevated Plus-Maze in Scopolamine-Induced Alzheimer’s Disease in Wistar Rats

On Day 29, there was no significant difference found in the transfer latency amongst the groups, whereas on day 30 the scopolamine-only group (27.91 ± 1.91) showed a significant (*p* < 0.001) increase in transfer latency as compared to the control group (12.61 ± 1.49), which indicated a loss of memory in the scopolamine-treated groups.

The SCO + memantine group (14.40 ± 2.42) showed a statistically significant (*p* < 0.001) decrease in transfer latency as compared to the scopolamine group (27.91 ± 1.91). SCO + CUR (200 mg/kg) + CoQ10 (200 mg/kg) (15.25 ± 1.22) showed the highest decrease in transfer latency as compared to the scopolamine-only group (27.91 ± 1.91) and no statistically significant difference in transfer latency existed between the standard (14.40 ± 2.42) and SCO + CUR (200 mg/kg) + CoQ10 (200 mg/kg) (15.25 ± 1.22), indicating that the combination of SCO + CUR (200 mg/kg) + CoQ10 (200 mg/kg) (15.25 ± 1.22) showed the highest results in improving memory in elevated plus maze.

Increase in transfer latency in SCO + CUR (100 mg/kg) (22.91 ± 1.29) and SCO + CoQ10 (200 mg/kg) (19.98 ± 2.96) was highly significant (*p* < 0.001) and SCO + CUR (200 mg/kg) (17.8 ± 2.55) was less significant (*p* < 0.050 when compared to SCO + Memantine group (14.40 ± 2.40), indicating relatively lesser improvement in memory in elevated plus maze (Figure 2).

In SCO +CUR (200 mg/kg) +CoQ_10_ (200 mg/kg) (15.25 ± 1.22), there was decrease (*p* < 0.001) in transfer latency as compared to SCO + CoQ_10_ (200 mg/kg) (19.98 ± 2.96) indicating strong correlating effect of combination of SCO + CUR (200 mg/kg) + CoQ_10_ (200 mg/kg).

#### 4.1.2. Effect of Curcumin and CoQ10 on the Time Taken to Reach the Platform in the Morris Water Maze in Scopolamine-Induced Alzheimer’s Disease in Wistar Rats

On Day 29, the scopolamine-only group (70.55 ± 2.2) showed a significant (*p* < 0.001) increase in the time taken to reach the platform as compared to the control group (50.16 ± 2.09), which indicated a loss of memory in the scopolamine-treated groups.

The SCO + memantine group (54.28 ± 4.95) showed a statistically significant (*p* < 0.001) decrease in transfer latency as compared to the scopolamine group (70.55 ± 5.69). SCO + CUR (200 mg/kg) +CoQ10 (200 mg/kg) (58.01 ± 3.36) showed the highest decrease in the time taken to reach the platform as compared to the scopolamine-only group (54.28 ± 4.95) and no statistically significant difference in the time taken to reach the platform existed between the standard (54.28 ± 4.95) and SCO + CUR (200 mg/kg) + CoQ10 (200 mg/kg) (58.01 ± 3.36) groups, indicating that the combination of SCO + CUR (200 mg/kg) + CoQ10 (200 mg/kg) (58.01 ± 3.36) showed great results in improving memory in the Morris water maze.

In SCO + CUR (200 mg/kg) + CoQ10 (200 mg/kg) (58.01 ± 3.36), there was decrease (*p* < 0.001) in the time taken to reach the platform as compared to SCO + CUR (200 mg/kg) (62.96 ± 6.94) as well as SCO + CoQ10200 mg/kg (63.4 ± 4.58), indicating a strong additive effect of the combination of SCO + CUR (200 mg/kg) + CoQ10 (200 mg/kg) (58.01 ± 3.36).

On Day 30, the scopolamine-only group (56.26 ± 2.2) showed a significant (*p* < 0.001) increase in the time taken to reach the platform as compared to the control group (34.65 ± 3.97), which indicated a loss of memory in the scopolamine-treated groups.

The SCO + memantine group (38.43 ± 3.09) showed a statistically significant (*p* < 0.001) decrease in transfer latency as compared to the scopolamine group (34.65 ± 3.97). SCO + CUR (200 mg/kg) +CoQ10 (200 mg/kg) (40.75 ± 3.91) showed the highest decrease in the time taken to reach the platform as compared to the scopolamine-only group (56.26 ± 2.2) and no statistically significant difference in the time taken to reach the platform existed between the standard (38.43 ± 3.09) and SCO + CUR (200 mg/kg) + CoQ10 (200 mg/kg) (40.75 ± 3.91), indicating that the combination of SCO + CUR (200 mg/kg) + CoQ10 (200 mg/kg) (40.75 ± 3.91) showed positive results in improving memory in the Morris water maze (Figure 3).

In SCO + CUR (200 mg/kg) + CoQ10 (200 mg/kg) (40.75 ± 3.91), there was a decrease (*p* < 0.001) in the time taken to reach the platform as compared to SCO + CUR 200 mg/kg (44.65 ± 5.97) as well as SCO +CoQ10 (200 mg/kg) (47.7 ± 7.95), indicating a strong additive effect of the combination of SCO + CUR (200 mg/kg) + CoQ10 (200 mg/kg) (40.75 ± 3.91).

#### 4.1.3. Effect of Curcumin and Coenzyme Q10 on Percentage Alternations in Y-Maze Test in Scopolamine-Induced Alzheimer’s Disease in Wistar Rats

The scopolamine-only group (48.19 ± 2.55) showed a significant (*p* < 0.001) decrease in percentage alternations as compared to the control group (62.94 ± 3.84), which indicated a loss of memory in the scopolamine-treated groups.

The SCO + memantine group (60.48 ± 3.49) showed a statistically significant (*p* < 0.001) increase in percentage alternations as compared to the scopolamine group (48.19 ± 2.55). SCO + CUR 200 mg/kg + CoQ10 200 mg/kg (59.45 ± 2.88) showed the highest increase in percentage alternations as compared to the scopolamine-only group (48.19 ± 2.55), and no statistically significant difference in percentage alternations existed between the standard (60.48 ± 3.49) and SCO + CUR 200 mg/kg + CoQ10200 mg/kg (59.45 ± 2.88) groups, indicating that the combination of SCO + CUR 200 mg/kg + CoQ10200 mg/kg (59.45 ± 2.88) showed great results in improving memory in the Y maze (Figure 4).

In SCO + CUR 200 mg/kg + CoQ10200 mg/kg (59.45 ± 2.88), there was an increase (*p* < 0.05) in percentage alternations as compared to SCO + CoQ10 200 mg/kg (54.98 ± 3.61) as well as SCO +CoQ10 200 mg/kg (54.98 ± 3.61), indicating a strong additive effect of the combination of SCO + CUR 200 mg/kg + CoQ10 200 mg/kg (59.45 ± 2.88) on memory.

### 4.2. Biochemical Parameters

#### 4.2.1. Effect of Curcumin and Coenzyme Q10 on Acetylcholinesterase (AChE) Level in Brain Homogenate in Scopolamine-Induced Alzheimer’s Disease in Wistar Rats

The Scopolamine-Only Group (922.45 ± 26.99) had a Significantly (*p* < 0.001) Higher Total AChE Level as Compared to the Control Group (474.4 ± 13.70), which Indicated less Acetylcholine-Like Activity in the Scopolamine-Only Group.

The SCO + memantine group (870.58 ± 34.95) showed a statistically significant (*p* < 0.001) decrease in AChE level as compared to the scopolamine group (922.45 ± 26.99). SCO + CUR 200 mg/kg + CoQ10200 mg/kg (738.94 ± 24.86) showed the highest decrease in AChE level as compared to the scopolamine-only group (922.45 ± 26.99), along with a highly statistically significant (*p* < 0.001) difference in AChE level existing between the standard (870.58 ± 34.95) and SCO + CUR 200 mg/kg + CoQ10200 mg/kg (738.94 ± 24.86) groups, indicating that the combination of SCO + CUR 200 mg/kg + CoQ10 200 mg/kg was most effective amongst these groups in decreasing AChE level in brain homogenates (Figure 5).

There was a highly significant (*p* < 0.001) difference between the SCO + CUR (200 mg/kg) (779.85 ± 33.16) and SCO + CUR 200 mg/kg + CoQ10 200 mg/kg (738.94 ± 24.86) groups and SCO + CoQ10 (200 mg/kg) (789.2 ± 27.9) and SCO + CUR 200 mg/kg + CoQ10200 mg/kg (738.94 ± 24.86), with SCO + CUR 200 mg/kg + CoQ10200 mg/kg (738.94 ± 24.86) showing the highest decrease in AChE level.

#### 4.2.2. Effect of Curcumin and Coenzyme Q10 on Superoxide Dismutase (SOD) Level in Brain Homogenate in Scopolamine-Induced Alzheimer’s Disease in Wistar Rats

The scopolamine-only group (11.58 ± 0.56) had a significantly (*p* < 0.001) lower total SOD level as compared to the control group (26.1 ± 1.21), which indicated more oxidative stress in the scopolamine-only group.

The SCO + memantine group (23.94 ± 0.45) showed a statistically significant (*p* < 0.001) increase in SOD level as compared to the scopolamine group (11.58 ± 0.56). SCO + CUR 200 mg/kg +CoQ10 200 mg/kg (28.22 ± 0.57) showed the highest increase in SOD level as compared to the scopolamine-only group (11.58 ± 0.56) and even surpassing the standard (23.94 ± 0.45) group with a statistically significant (*p* < 0.001) increase in SOD level, indicating that the combination of SCO + CUR 200 mg/kg + CoQ10 200 mg/kg (28.22 ± 0.57) was most effective amongst these groups in improving SOD level in brain homogenates and acting as a powerful antioxidant (Figure 6).

There was a highly significant (*p* < 0.001) difference between SCO + CUR 200 mg/kg (23.58 ± 2.14) and SCO + CUR 200 mg/kg +CoQ10 200 mg/kg (28.22 ± 0.57) as well as SCO + CoQ10200 mg/kg (22.23 ± 1.11) and SCO + CUR 200 mg/kg +CoQ10200 mg/kg (28.22 ± 0.57), with the SCO + CUR 200 mg/kg +CoQ10 200 mg/kg (28.22 ± 0.57) group showing a higher increase in SOD level indicating the role of SCO + CUR 200 mg/kg +CoQ10 200 mg/kg (28.22 ± 0.57) in reducing oxidative stress.

#### 4.2.3. Effect of Curcumin and Coenzyme Q10 on Tumour Necrosis Factor α (TNFα) Level in Brain Homogenate in Scopolamine-Induced Alzheimer’s Disease in Wistar Rats

The Scopolamine-Only Group (279.84 ± 11.57) had a Significantly (*p* < 0.001) Higher Total TNFα Level as Compared to the Control Group (35.94 ± 4.51), which Indicated More Inflammation in the Scopolamine-Only Group

The SCO + memantine group (124.11 ± 8.61) showed a statistically significant (*p* < 0.001) decrease in TNFα level as compared to the scopolamine group (279.84 ± 11.57). SCO + CUR 200 mg/kg +CoQ10 200 mg/kg (144.87 ± 6.84) showed the highest decrease in TNFα level as compared to the scopolamine-only group (279.84 ± 11.57), along with a highly statistically significant (*p* < 0.001) difference in TNFα level existing between the standard (124.11 ± 8.61) and SCO + CUR 200 mg/kg +CoQ10 200 mg/kg (144.87 ± 6.84) groups, indicating that the combination of SCO + CUR 200 mg/kg + CoQ10 200 mg/kg (144.87 ± 6.84) was most effective amongst these groups in decreasing TNFα level in brain homogenates (Figure 7).

There was a significant (*p* < 0.05) difference between SCO + CUR 200 mg/kg (169.80 ± 9.87) and SCO + CUR 200 mg/kg +CoQ10 200 mg/kg (144.87 ± 6.84) and between SCO + CoQ10 200 mg/kg (169.80 ± 9.87) and SCO + CUR 200 mg/kg +CoQ10 200 mg/kg (144.87 ± 6.84) with SCO + CUR 200 mg/kg +CoQ10 200 mg/kg (144.87 ± 6.84) showing a higher decrease in TNFα level.

#### 4.2.4. Effect of Curcumin and Coenzyme Q10 on Aβ42 Level in Brain Homogenate in Scopolamine-Induced Alzheimer’s Disease in Wistar Rats

Aβ42 was not significantly elevated in any of the groups, including drug-treated groups, and this can be regarded as scientific evidence that Aβ42 is not formed in the scopolamine-induced Alzheimer’s disease model of Wistar rats.

## 5. Discussion

The present study has been designed to evaluate the cognition (learning and memory enhancement) effect of curcumin and coenzyme Q10 alone and in combination. These effects were compared with the standard drug Memantine for learning and memory.

We used a purified form of curcumin (powder), which is an active compound of *Curcuma longa*. We chose the oral route for giving the test drug, as it is a usual and convenient method of taking a drug and ensures better compliance.

As mentioned in Materials and Methods, there were a total of seven groups of Wistar rats, and Alzheimer’s disease was induced in six groups out of them. The disease was induced by giving scopolamine injection intraperitoneally in all groups except the control group.

As in other research, amnesia is induced using the scopolamine model (2.5 mg/kg) [31]. It has a central action and is a muscarinic cholinergic antagonist. It causes forgetfulness because cholinergic neurotransmission is hindered. It induces central cholinergic blockade when given systemically, which affects mice’s capacity to retain information, pay attention, and create new memories. Scopolamine was therefore used in our investigation due to this mechanism.

Three behavioural tests were used in this work to examine the behavioural changes in experimental rats following scopolamine administration: the Y maze, the Morris water maze (MWM), and the elevated plus maze (EPM). To assess the impact of curcumin and coenzyme Q10, the following parameters were observed: percentage alternation in the Y maze, time to reach the platform in MWM, and transfer delay in EPM. In order to analyse them as biomarkers for learning and memory assessment, we also assessed the levels of acetylcholinesterase, TNF, Superoxide dismutase (SOD), and Amyloid-42 using rat brain homogenates [32].

The changes observed in behaviour during the course of the experimental period can be attributed to the impact of scopolamine on the brains of Wistar rats. Treatment of rats with curcumin (herbal drug) and in combination with coenzyme Q10 improved the behaviour (i.e., increase in percentage alterations in Y maze, decrease in time taken to reach the platform in Morris water maze, decrease in transfer latency in elevated plus maze) and also decreased anxiety associated with scopolamine (2.5 mg/kg). The result of the combination (Curcumin—200 mg/kg + CoQ10200 mg/kg) treatment was found to be much better.

In this study, in order to investigate oxidative stress in experimental rats after giving scopolamine, the biochemical test of SOD level was observed. Oxidative stress marker level (SOD) was improved in the group receiving curcumin (200 mg/kg) alone and in the group receiving the combination of curcumin (200 mg/kg) and coenzyme Q10 (200 mg/kg). The results of the combination (Curcumin—200 mg/kg + CoQ10200 mg/kg) treatment were found to be much better. This further strengthens the possibility of an antioxidant role of curcumin and coenzyme Q10. Acetylcholine (Ach) has a crucial role in the peripheral and central nervous systems. Cholinergic neurons located in the basal forebrain, including the neurons that form the nucleus basalis of Meynert, are severely lost in Alzheimer’s disease (AD).

In this study, to investigate the level of acetylcholine in experimental rats after giving scopolamine, an AchE level was observed, which showed the maximum improvement in the group receiving the combination of curcumin (200 mg/kg) and coenzyme Q10 (200 mg/kg).

In this study, to investigate inflammation in experimental rats after giving scopolamine, TNFα level was observed and Aβ42 levels were also observed with TNFα levels being decreased in the group receiving a combination of curcumin (200 mg/kg) and coenzyme Q10 (200 mg/kg), as well as in the group receiving Curcumin 200 mg/kg and CoQ10 200 mg/kg alone. The result of the combination (Curcumin—200 mg/kg + CoQ10—200 mg/kg) treatment is much better and was comparable to standard drug treatment.

This study demonstrated that curcumin is capable on its own to improve the behavioural, biochemical, and inflammatory parameters of the scopolamine-induced AD model. However, when combined with coenzyme Q10 (200 mg/kg), the results are accentuated even further in the positive direction as was evident with a decrease in transfer latency in the elevated plus maze, a decrease in the time taken to reach the platform in the Morris water maze, a decrease in AchE and TNFα, and an increase in SOD levels. Based on different experimental results, the present investigation suggested that the curcumin (200 mg/kg) and coenzyme Q10 (200 mg/kg) combination displayed significantly better results in opposing the cognitive, biochemical, and inflammatory defects created by the scopolamine-induced Alzheimer’s disease model of Wistar rats.

### 5.1. The Following Studies Are in Accordance with Our Present Study

The preventive and therapeutic effects of 200 mg of curcumin on SDAT (Senile Dementia of Alzheimer’s Type) were examined by Agrawal et al. in 2010 [33]. Oral doses of curcumin were given either six days after the onset of the disease or fourteen days prior to it. In both circumstances, curcumin administration improved memory function in the Morris water maze (MWM). Additionally, following treatment with curcumin (200 mg/kg), levels of oxidative stress, acetylcholine, and insulin were recovered [33].

The comparative protective effects of curcumin, memantine, and diclofenac against scopolamine-induced memory impairment were investigated by Elham H.A. Ali et al. (2011) [34] in a group of male and female rats that had been administered one of these substances for 15 days previous to receiving a single dose of the medication. Despite using both medications in lower doses, curcumin had a higher protective impact against memory loss than diclofenac or memantine. Gender differences were discovered. Before administering a single dose of scopolamine, a group of male and female rats were given one of these drugs for 15 days. Despite using both medications in lower doses, curcumin had a higher protective impact against memory loss than diclofenac or memantine. Gender differences were discovered.

Zhang et al. (2015) [35] assessed the protective effects of curcumin at doses of 50, 100, and 200 mg/kg, i.p. on Aβ-42 mice that had received intraventricular injections. In acute therapy, curcumin had no beneficial effects. However, compared to placebo, prolonged therapy (7 day administration) with 200 mg/kg curcumin reduced cognitive deficits, demonstrated by the Y maze and MWM. ICV-STZ-infused rats’ short-term spatial memory did not improve after extended oral administration of curcumin at dosages of 25, 50, and 100 mg/kg, according to Bassani et al. (2017) [36].

The expression of nuclear factor-B (NF-B), tumour necrosis factor (TNF), and interleukin-6 (IL-6), which is increased by the medication 1-methyl-4-phenyl-1,2,3,6-tetrahydropyridine, has been demonstrated to be inhibited by coenzyme Q10. This has been linked to pleiotropic and anti-inflammatory effects (MPTP). According to studies on coenzyme Q10′s role, improved mitochondrial activity lowers oxidative damage, which lowers age-related cognitive decline [37,38]. Age-related mitochondrial failure has been demonstrated to be reversible, and oxidative stress and mitochondrial dysfunction have both been linked to cognitive decline and the emergence of certain neurodegenerative diseases [39].

As of date, none of the studies have observed the effect of curcumin and coenzyme Q10 in combination for learning and memory enhancement (cognition) in an animal model of Alzheimer’s disease; hence, this is a combination that should be explored further.

### 5.2. Possible Explanations and the Mechanism for the Effect on Learning and Memory Enhancement That We Obtained in Our Study with Curcumin Are as Follows

Curcumin’s antioxidant and anti-inflammatory properties account for most of its beneficial benefits on the variety of illnesses described in Jurenka JS’ review. (2009) [40].

Curcumin has been shown to ameliorate systemic signs of oxidative stress. There is evidence that it can increase the serum activity of antioxidants such as superoxide dismutase (SOD) [41]. Free radicals are impacted by curcumin through a number of methods. It may scavenge different kinds of free radicals, including reactive oxygen and nitrogen species, as well as block ROS-generating enzymes including lipoxygenase/cyclooxygenase and xanthine hydrogenase/oxidase (ROS and RNS, respectively). The activity of the enzymes GSH, catalase, and SOD, which are involved in the neutralisation of free radicals, can also be modulated [42,43].

A few reactive oxygen/nitrogen species can also set off a chain of intracellular signals that encourage the expression of genes that cause inflammation. Inflammation has been related to several chronic illnesses and ailments in their early stages of development [44]. One of these illnesses is Alzheimer’s disease (AD).

Tumour necrosis factor (TNF-), a key mediator of inflammation in the majority of diseases, activates nuclear factor (NF)-B, a transcription factor. NF-B also regulates the expression of TNF-, despite the fact that TNF- is regarded to be the most potent NF-B activator. As was also seen in the current study, curcumin has been shown to suppress NF-B activation, which is brought on by a number of inflammatory stimuli [45,46,47].

### 5.3. Possible Explanations and the Mechanism for the Effect on Learning and Memory Enhancement That We Obtained in our Study with CoQ10 Are as follows

Coenzyme Q10 (CoQ10) can decrease oxidative stress, hyperglycemia, and inflammatory markers, and improve vascular function. Coenzyme Q10 supplementation influences the cholinergic system and protects cholinergic neurons in patients with Alzheimer’s disease [48].

The results of the current investigation could shed light on coenzyme Q10’s efficacy in enhancing cognitive function in scopolamine-induced amnesic rats. In various animal experiments investigating the function of oxidative stress in neurodegenerative disorders, oxidative stress indicators and antioxidant concentrations were assessed. (ND) [49].

The use of herbal medicines as complementary and alternative medicines (CAM), particularly for neurological and mental illnesses as well as for physical wellbeing, is on the rise today all over the world. A protracted course of conventional standard treatment may be harmful and cause major adverse medication reactions. Traditional treatments, nutritional therapy, and herbal products are safer and more dependable. We attempted to combine curcumin and coenzyme Q10 in our study.

## 6. Conclusions

Given the results that were obtained from the study of pharmacological and biochemical/molecular parameters in the present study, the following conclusion may be drawn regarding the potential effectiveness of test drugs. Curcumin has a learning and memory enhancement effect and a higher dose (200 mg/kg b.w.s p.o.) is comparatively more effective than a lower dose (100 mg/kg b.w. p.o.) and it has shown a dose–response effect. Coenzyme Q10 (200 mg/kg b.w. p.o.) also has learning and memory enhancement effects which were comparable to the effect of a higher dose of curcumin The effects of the combination of curcumin and coenzyme Q10 on learning and memory were significant and greater than when both drugs were given alone and were comparable to Memantine.

The results of the present study are encouraging and may reveal the importance of curcumin and coenzyme Q10 herbal drugs and nutrients in impaired cognition states. As a result, curcumin, coenzyme Q10, and their combination, which has demonstrated the greatest benefit, may be effective in the management of learning- and memory-impaired states as an alternative, supplemental, or even preventive medication. These medications, alone or in combination, can be taken by patients of any age for a longer period of time without experiencing any negative side effects. To support our encouraging results, additional animal and clinical investigations are needed.

## Figures and Tables

**Figure 1 biomedicines-11-01422-f001:**
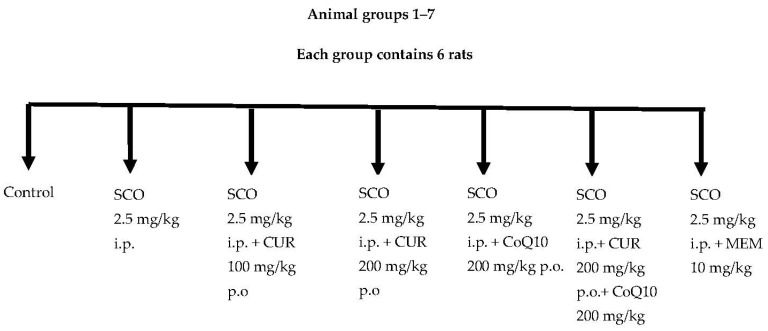
Animal groups.

**Figure 2 biomedicines-11-01422-f002:**
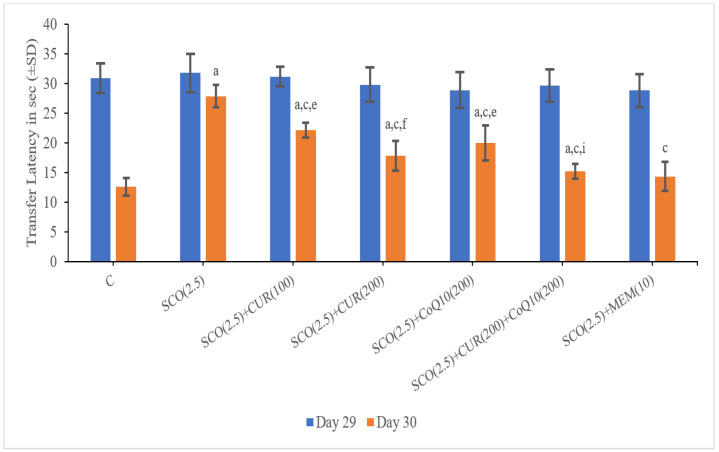
Representation of changes in transfer latency of curcumin and CoQ10 in elevated plus-maze in scopolamine-induced Alzheimer’s disease in Wistar rats. Statistically significant, a *p* < 0.001 as compared to control, c *p* < 0.001 as compared to SCO (2.5 mg/kg)), e *p* < 0.001 as compared to SCO + Memantine (10 mg/kg), f *p* < 0.05 as compared to SCO + Memantine (10 mg/kg), i *p* < 0.001 as compared to SCO + CoQ10 (200 mg/kg).

**Figure 3 biomedicines-11-01422-f003:**
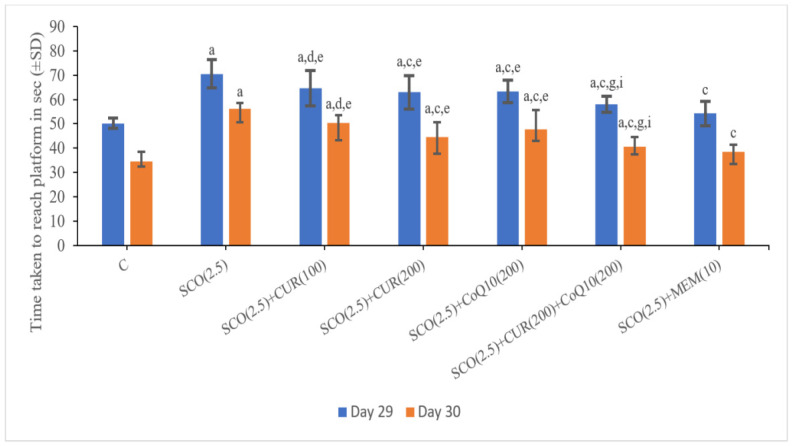
Representation of changes in time taken to reach the platform of curcumin and CoQ10 in Morris water maze in scopolamine-induced Alzheimer’s disease in Wistar rats. Statistically significant, a *p* < 0.001 as compared to control, c *p* < 0.001 as compared SCO (2.5 mg/kg), d *p* < 0.05 as compared to SCO (2.5 mg/kg), e *p* < 0.001 as compared to SCO + Memantine (10 mg/kg), g *p* < 0.001 as compared to SCO + CUR (200 mg/kg), i *p* < 0.001 as compared to SCO + CoQ10 (200 mg/kg).

**Figure 4 biomedicines-11-01422-f004:**
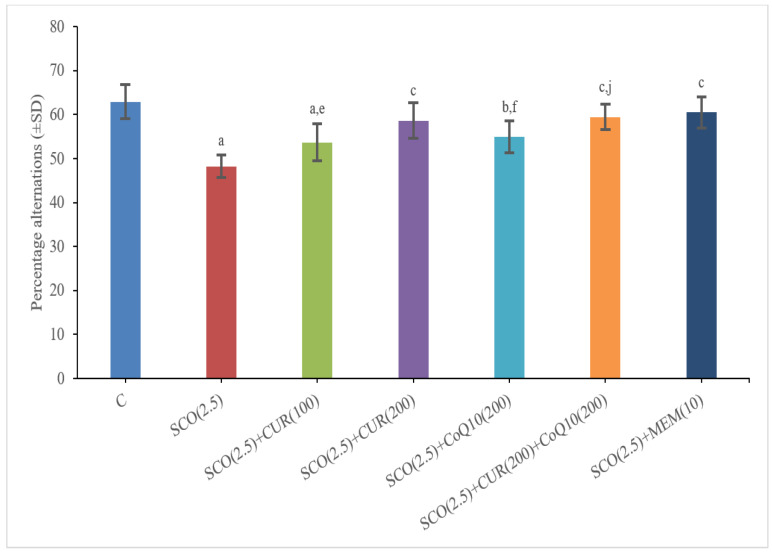
Representation of changes in percentage alternations of Curcumin and CoQ10 in Y-maze test in scopolamine-induced Alzheimer’s disease in Wistar rats. Statistically significant, a *p* < 0.001 as compared to control, b *p* < 0.05 as compared to control, c *p* < 0.001 as compared to SCO (2.5 mg/kg), e *p* < 0.001 as compared to SCO + Memantine (10 mg/kg, f *p* < 0.05 as compared to SCO + Memantine (10 mg/kg), j *p* < 0.05 as compared to SCO + CoQ10 (200 mg/kg).

**Figure 5 biomedicines-11-01422-f005:**
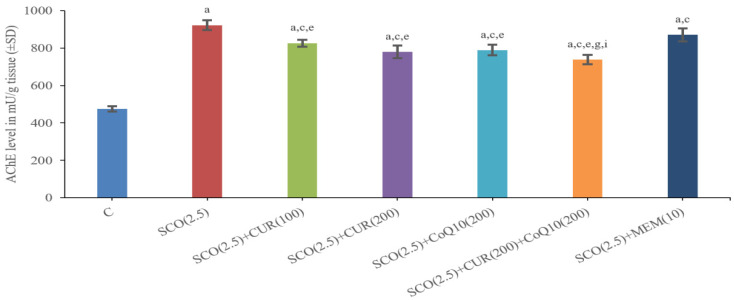
Representation of changes in acetylcholinesterase (AChE) of Curcumin and CoQ10 in scopolamine-induced Alzheimer’s disease in Wistar rats. Statistically significant, a *p* < 0.001 as compared to control, c *p* < 0.001 as compared to SCO (2.5 mg/kg), e *p* < 0.001 as compared to SCO + Memantine (10 mg/kg), g *p* < 0.001 as compared to SCO + CUR (200 mg/kg), i *p* < 0.001 as compared to SCO + CoQ10 (200 mg/kg).

**Figure 6 biomedicines-11-01422-f006:**
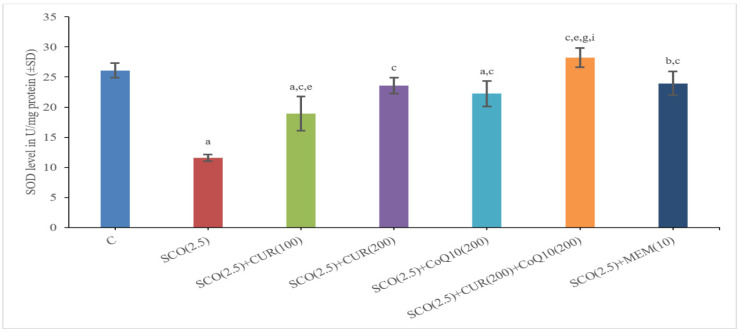
Representation of changes in superoxide dismutase (SOD) of Curcumin and CoQ10.in scopolamine-induced Alzheimer’s disease in Wistar rats. Statistically significant, a *p* < 0.001 as compared to control, b *p* < 0.05 as compared to control, c *p* < 0.001 as compared to SCO (2.5 mg/kg), e *p* < 0.001 as compared to SCO + Memantine (10 mg/kg), g *p* < 0.001 as compared to SCO + CUR (200 mg/kg), i *p* < 0.001 as compared to SCO + CoQ10 (200 mg/kg).

**Figure 7 biomedicines-11-01422-f007:**
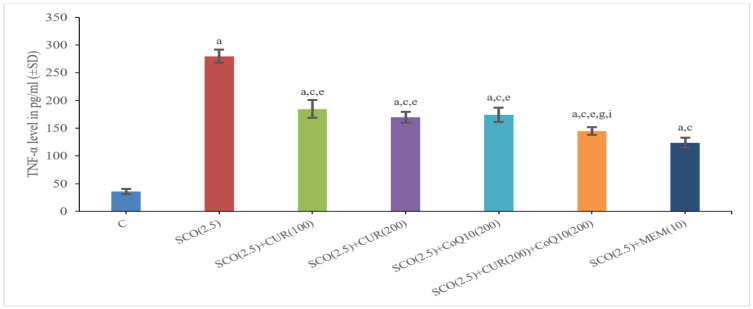
Representation of changes in TNFα of curcumin and CoQ10 in scopolamine-induced Alzheimer’s disease in Wistar rats. Statistically significant, a *p* < 0.001 as compared to control, c *p* < 0.001 as compared to SCO (2.5 mg/kg), e *p* < 0.001 as compared to SCO + Memantine (10 mg/kg), g *p* < 0.001 as compared to SCO + CUR (200 mg/kg), i *p* < 0.001 as compared to SCO + CoQ10 (200 mg/kg).

**Table 1 biomedicines-11-01422-t001:** Animal groups.

Group	Name	No. of Rats
1.	Control (Vehicle)	6
2.	SCO 2.5 mg/kg i.p.	6
3.	SCO 2.5 mg/kg i.p. + CUR 100 mg/kg p.o.	6
4.	SCO 2.5 mg/kg i.p. + CUR 200 mg/kg p.o.	6
5.	SCO 2.5 mg/kg i.p. + CoQ10 200 mg/kg p.o.	6
6.	SCO 2.5 mg/kg i.p.+ CUR 200 mg/kg p.o.+ CoQ10 200 mg/kg p.o.	6
7.	SCO 2.5 mg/kg i.p. + MEM 10 mg/kg i.p.	6

## Data Availability

No data available.

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
