# Peer review of "Effect of Curcumin and Coenzyme Q10 Alone and in Combination on Learning and Memory in an Animal Model of Alzheimer’s Disease"

_biomedicines, 2023, doi:10.3390/biomedicines11051422_

Round 1
Reviewer 1 Report
The authors of the manuscript presented their study evaluating the effects of curcumin and/or coenzyme Q10 on learning and memory in an animal model of Alzheimer's disease compared to memantine. However, in its current form, the manuscript is not suitable for publication in Biomedicine. To improve the manuscript, the authors should conduct a thorough review of recent scientific literature regarding the use of curcumin and CoQ10 in the prevention or treatment of Alzheimer's disease symptoms. The literature cited by the authors is outdated, and a more comprehensive review would be more informative.
Additionally, the authors should clearly justify why they chose to analyze these compounds. The rationale behind selecting these compounds and their mode of action in the context of Alzheimer's disease should be more detailed.
Moreover, similar studies were previously described e.g. in 2011 (https://doi.org/10.1016/j.fitote.2011.01.016), and the authors should acknowledge and cite these studies to provide context and avoid redundancy.
In my opinion, it is also important to refer to the bioavailability of curcumin and CoQ10. The bioavailability of these compounds can impact their efficacy and should be taken into account in the analysis.
Furthermore, the results presented in the figures are not very clear. The authors should improve the figures to ensure that the results are easily understandable for readers.
It is also worth noting that the explanations of the mechanism of action of curcumin and CoQ10 in nervous system diseases have already been described in earlier publications and do not constitute an element of novelty. Therefore, the authors should focus on providing more original insights and interpretations of their results.
Moreover, the authors should specify possible limitations of the use of their chosen compounds in Alzheimer's disease, such as interactions, to give a more balanced view of their potential applications.
Finally, the manuscript requires editorial refinement to improve clarity, coherence, and readability. In conclusion, addressing these issues will improve the manuscript's quality and make it more suitable for publication in Biomedicine.
Author Response
Review report -1
- English language & grammer has been corrected & updated.
- To improve the manuscript, the authors should conduct a thorough review of recent scientific literature regarding the use of curcumin and CoQ10 in the prevention or treatment of Alzheimer's disease symptoms. The literature cited by the authors is outdated, and a more comprehensive review would be more informative.-
- Literature has been updated. (Page:2&3)
- Additionally, the authors should clearly justify why they chose to analyze these compounds. The rationale behind selecting these compounds and their mode of action in the context of Alzheimer's disease should be more detailed-
-Justification & mode of action has been updated. (Page:3)
- Moreover, similar studies were previously described e.g. in 2011 (https://doi.org/10.1016/j.fitote.2011.01.016), and the authors should acknowledge and cite these studies to provide context and avoid redundancy-
-This study has been included as directed by the reviewer. (Page:14)
- In my opinion, it is also important to refer to the bioavailability of curcumin and CoQ10. The bioavailability of these compounds can impact their efficacy and should be taken into account in the analysis-
-Bioavailability of Curcumin has been added. (Page:3)
- The results presented in the figures are not very clear. The authors should improve the figures to ensure that the results are easily understandable for readers.-
-Fig. 1 has been revised along with inclusion of table-1 with addition of supplymentry legends showing significance in the figures has also been added. (Page:5,10,11,12 &13)
- It is also worth noting that the explanations of the mechanism of action of curcumin and CoQ10 in nervous system diseases have already been described in earlier publications and do not constitute an element of novelty. Therefore, the authors should focus on providing more original insights and interpretations of their results.-
-Explanation for the mechanism of action of Curcumin & CoQ10 has been updated. ( Page:2)
- The authors should specify possible limitations of the use of their chosen compounds in Alzheimer's disease, such as interactions, to give a more balanced view of their potential applications.-
-Limitations of the use of Curcumin has been included. (Page:3).
Reviewer 2 Report
Dear Authors The title must be reformulated, the abstract in the paragraph of Aim the english should be corrected with the text, the experimental designs should be in a table, fig 1 must be clear and original drawing by authors, Same fig 2 and 3 should be corrected The reference for example ref 13 and 14, the abbreviation of the journal is required Ref 33 need the abbreviation the page and the volume also...
Author Response
Review report -2
- The title must be reformulated, the abstract in the paragraph of Aim the english should be corrected with the text, the experimental designs should be in a table, fig 1 must be clear and original drawing by authors, Same fig 2 and 3 should be corrected The reference for example ref 13 and 14, the abbreviation of the journal is required Ref 33 need the abbreviation the page and the volume also.
- English language & grammer has been corrected & updated.
- Fig. 1 has been revised along with inclusion of table-1 with addition of supplymentry legends showing significance in the figures has also been added. (Page:5,10,11,12 &13)
& Ref.33 has been corrected & updated as Ref. 42.
All the comments made by Reviewers has been taken care of except the title which used to be self explanatory
Round 2
Reviewer 1 Report
The scientific value of the revised manuscript increased after the authors incorporated the suggested revisions. My comments on the submitted version are as follows:
1. The Latin names of plants should be in accordance with accepted standards - Line 131
2. The spelling of compounds should be standardized e.g. curcumin and coenzyme Q10 whether with a lowercase or uppercase letter in the middle of a sentence.
3. References should be standardized according to Biomedicine standards.
Author Response
The scientific value of the revised manuscript increased after the authors incorporated the suggested revisions. My comments on the submitted version are as follows:
- The Latin names of plants should be in accordance with accepted standards - Line 131
- The latin name of the plant has been corrected-line 131
- The spelling of compounds should be standardized e.g. curcumin and coenzyme Q10 whether with a lowercase or uppercase letter in the middle of a sentence.
- The spelling of compounds e.g. curcumin and coenzyme Q10 has been standardized
- References should be standardized according to Biomedicine standards.
- References has been standardized according to Biomedicine standards.
Reviewer 2 Report
All corrections done
Author Response
Answer have already been given to the queries of reviewer in our previous attachments